# Linking Entrepreneurial Innovation to Effectual Logic

Faiez Ghorbel [1], Wafik Hachicha [2,*], Younes Boujelbene [1] and Awad M. Aljuaid [2]

1 Department of Industrial Management, University of Sfax, Sfax 3029, Tunisia; faiez.ghorbel@isgis.usf.tn (F.G.); younes.boujelbene@fsegs.rnu.tn (Y.B.)

2 Department of Industrial Engineering, College of Engineering, Taif University, Taif 21944, Saudi Arabia; amjuaid@tu.edu.sa

* Correspondence: w.hashisha@tu.edu.sa or wafik.hachicha@isgis.usf.tn; Tel.: +966-53-194-0695

**Abstract:** The terms "innovation" and "effectuation" are frequently used but not in the same thought. In this order, publications linking innovation to effectuation are presented and discussed through a methodology based on the publish and perish tool. In the last two decades, effectuation has become an active criterion in entrepreneurship research. However, previous studies do not interconnect effectuation to the different innovation approaches. In order to overcome this gap, this paper focuses on studying innovation in an effectual context and linking different innovation approaches to effectual logic. Indeed, effectuation is a way of thinking that serves entrepreneurs in the processes of opportunity identification and new venture creation. Effectuation includes a set of decision-making principles expert entrepreneurs are observed to employ in situations of uncertainty (as defined in Society for Effectual Action). This article outlines the four most-studied innovation approaches from the date of their apparitions until January 2021: frugal innovation, disruptive innovation, lean start-up, and design thinking. In this context, effectuation as the essence of innovation must be clarified as a method that has similarities and differences with frugal and disruptive innovation, lean start-up, and design thinking. To validate the proposed theorical model, a bibliometrics tool, named "Harzing publish" or "perish", is used. The main finding of this research affirms that the two most linked innovation approaches to effectuation are "lean start-up" and "design thinking", compared to "frugal innovation" and "disruptive innovation". In an entrepreneurial innovation context, design thinking and lean start-up are flexible tools that can stimulate and validate the effectual cycle.

**Keywords:** effectuation; frugal innovation; disruptive innovation; lean start-up; design thinking; literature survey

## 1. Introduction

### 1.1. Research Motivation

Innovation is like fire: rapid, fluctuating, and variable. Research in innovation has similarly sought to identify the magic substance that feeds the value creation, regional development, and sustainability of its effectuation. The effectual cycle evolved from the practice of entrepreneurial innovation. In this order, the different methods of innovation, typically lean methods based on fostering innovation, rapid iteration, user-centered approach, and test prototypes, can be considered vital to the validation of business models, entrepreneurial cycle, product, opportunities, and market co-creation.

Researchers would disagree that effectuation and innovation are the keys triggers of the entrepreneurial activities. However, innovation and effectuation have been treated with different conceptions for many years. An effectual entrepreneur depends on his action in the presence of innovative ideas. Innovation is a historical phenomenon, which is the subject of practices developed in different study periods. Nowadays, approaches and notions, such as design thinking, lean start-up (appeared in 2012), frugal innovation (since 2013), and disruptive innovation (first coined by Harvard professor Clayton M. Christensen), are of great importance. This study includes the extension in time of those practices.

Therefore, the principles of effectuation theory include many kinds and conceptions of entrepreneurial innovation. This intersection can be valuable for entrepreneurial firms to create new products, firms, and markets.

In the last two decades, effectuation has become an active criterion in entrepreneurship research. However, previous studies do not interconnect effectuation to the different types and approaches of innovation. In order to overcome this gap, this research suggests a method to link effectuation to the four previously mentioned types and approaches of innovation. The two main research questions studied in this paper are about which innovation type is more connected to effectuation and uses a literature survey to confirm the obtained results. The methodology of this article includes three major steps: (1) reviewing the definition, the origin, and the major innovation characteristics; (2) exploring the effectuation concept and linking innovation to entrepreneurship; and (3) linking effectuation as an entrepreneurship theory with the four most-studied innovation approaches from the date of their apparitions until January 2021.

### 1.2. Effectuation

Effectuation is acclaimed as a rigorous framework for understanding the creation and growth of new organizations and markets [1]. The key differences between effectuation and causation are the starting point of the entrepreneurial process. Sarasvathy [2] argued that the causation model starts with goals. In the effectuation context, entrepreneurs start with the available means and try to imagine the possible actions [3]. Causal problems are the problems of a decision; effectual problems are the problems of a design [1]. In her groundbreaking article, Sarasvathy [2] discussed causation and effectuation as different approaches to the venture creation process. The expert frame is called 'effectual' because it proceeds outward from the means and cause to new effects and unanticipated ends [4].

Sarasvathy [2] compared and contrasted causation and effectuation models by applying the analogy of a chef assigned to the task of cooking dinner. In the causal case, the chef selects a menu, comes up with good recipes for each item on the menu, shops for the necessary ingredients, arranges the proper implements and appliances, and then cooks the meal. The causal process starts with selecting a menu as the goal and finding effective ways to achieve it. In the effectual case, the chef starts with looking through the kitchen cupboards for ingredients and utensils, then designs possible menus based on those ingredients and utensils. In fact, the menu often emerges while preparing the meal. The effectual chef starts with a given kitchen, and designs possible, sometimes unintended, and even entirely original meals with their content [2]. The Figure 1 shows the effectual cycle and its principles.

The theoretical basis of effectuation is based on five principles and a dynamic cycle [5]. Effectuation is action oriented. The effectual entrepreneur starts with the available means. This principle includes who I am, what I know, and whom I know. Then, the entrepreneurs imagine possibilities that originate from their means. In addition, effectual entrepreneurs set what they can afford to lose at each step. Therefore, the entrepreneur tries to introduce a partnership with self-selecting stakeholders. The partnership is essential for an entrepreneur to interpret news and surprises as potential signs to create opportunities and markets. Finally, the entrepreneur controls this environment to cope with uncertainty influences. The observer of all those ingredients from the dynamic effectual cycle notes this finding: do not wait for an opportunity but create it.

Effectuation principles are the foundation of entrepreneurial action. Theoretically, effectuation is noted by pioneers as action oriented. Sarasvathy [2] highlighted five principles used by expert entrepreneurs in their decisions. Those principles are the (1) bird in hand, (2) the affordable loss, (3) crazy patchwork, (4) the lemonade, and (5) pilot in the plane.

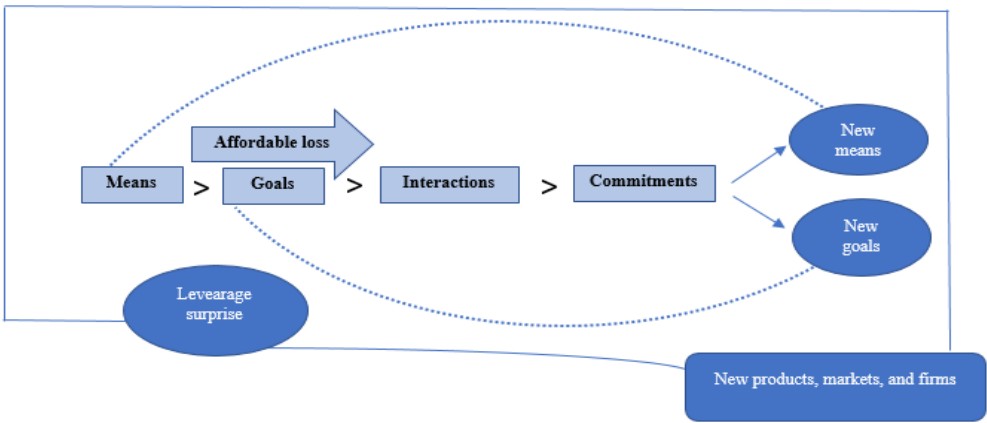

**Figure 1.** Effectual cycle and principles.

In effectuation logic, entrepreneurs start with their means, which can be grouped in three categories: who I am (personality), what I know (expertise), and who I know (social network). Based on the combinations between these means, entrepreneurs imagine possibilities and take action [6,7].

Covin et al. [8]. stated that in case of affordable loss, entrepreneurs base their approach on cost control rather than on estimable incomes. The idea is the following: By the commitment to an entrepreneurial project, it is easy to limit its involvement in terms of cost. Affordable loss drives entrepreneur decisions about which venture to start. In this case, the prediction is rejected by the entrepreneur's action.

An effectual process does not evolve without the selection and the commitment of different stakeholders [9]. The "crazy patchwork" principle emanates from this idea. The chain of commitments launched at the start of the venture has important impacts. It increases the available means for entrepreneurs and supports them in an emerging approach [10].

In addition to the previous point, the project goals are also identified by the stakeholders who are involved in it. This naturally does not mean that you have to change your plans to any customer commitment [11].

Wiltbank et al. [12] defined effectuation as the movement from a logic of prediction (try to predict the market) to control logic (invent it). The control logic also means that in the entrepreneurial process, it is the action that is preferred for analysis. The action is the source of learning but also a way for environmental change. Action is a source of novelty.

*1.3. Innovation*

The word 'innovation emanates from the Latin word "innovare", which means 'doing something new. Innovation is a complex and a dynamic process. It is, typically, the road from ideas and opportunities to market. In order to define the innovation, a historical examination of the current debates and contributions is needed.

Schumpeter [13] depicted innovation as a dynamic process that causes transformation of social, institutional, and economic structures [14]. Schumpeter [12] defined innovation as *"The commercial or industrial application of something new–a new product, process or method of industrial production; a new market or source of supply; a new form of commercial, business or financial organization"* [15].

Schumpeter's theory of economic development is interconnected to the power effects of innovation and entrepreneurship. Actually, Schumpeterian innovation categories include:

- Product innovation—new or improved products (introducing a product that it is not previously available or experimented by users);
- Production innovation—new methods for turning inputs into outputs (integrated new method or conception in the production process);

- Market innovation—opening and developing new markets (new market creation);
- Supply innovation—new sources and methods of supply (development of available resources: increasing resource quality or decreasing resource costs);
- Organizational innovation—new ways of organizing business and work.

In addition, Mitra [16] mantained that innovation and entrepreneurship are seen as clearly interrelated as the role of 'the' entrepreneur in Schumpeter can only be understood if it is placed against the background of his theory of innovation. In the Schumpeterian conception, the entrepreneur is the principal actor of economic dynamics. Schumpeter [12] saw innovation as the creation of "new combinations". Moreover, as mentioned in Figure 2, Miller [17] described firm-level entrepreneurship as a multidimensional concept encompassing a firm's innovative action, proactiveness, and risk taking.

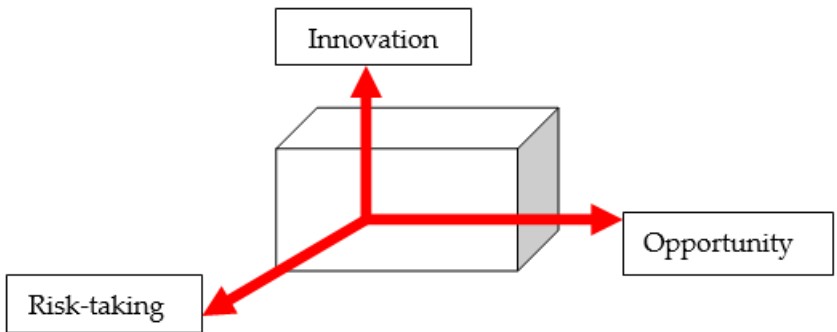

**Figure 2.** Firm-level entrepreneurship in Miller's study [16].

Innovation and entrepreneurship are generally viewed as interconnected concepts. They are seen as integrated components of the economic development and industrial renewing [18]. Patricio et al. [19] added that innovation and entrepreneurship have been treated within different scientific foundations. Drucker [20] said that innovation is the specific tool of entrepreneurs, the means by which they exploit change as an opportunity for a different business or a different service. Gojny-Zbierowska and Zbierowski [21] suggested that improvisation might be seen as a method of responsible innovation in organizations, due to its potential to be more responsive and enable bottom-up initiative.

Moreover, entrepreneurs and innovators are creative in diverse areas, such as design, science, technology, the arts, and organizational development, and they work for many different types of organizations [22]. The effort of entrepreneurs is oriented to adapt and transform opportunities into successful innovation [23]. The ideas of adaptation and transformation build the key idea of effectuation theory. Furthermore, entrepreneurs transform uncertainty into opportunities based on the expertise effect and a powerful and efficient innovation approach [2].

As mentioned above, many innovation classifications have been proposed in the literature. However, not all innovation types can be related to effectuation. In this research, four main innovation types are considered, which are theoretically the most linked to effectuation principles, include the following: design thinking, lean start-up, frugal innovation, and disruptive innovation (see Figure 3).

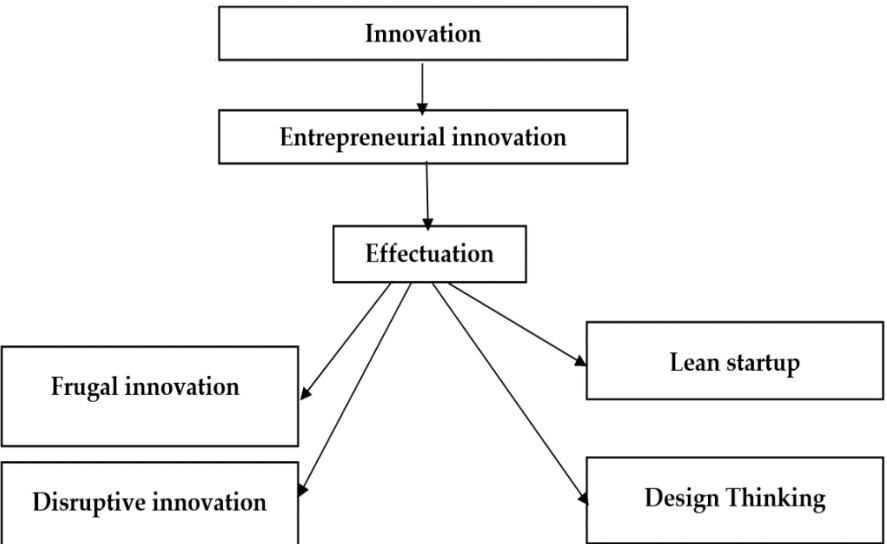

**Figure 3.** The proposed framework.

## 2. Materials and Methods

### 2.1. Framework Development and Research Methodology

The cornerstone of performance and effective innovation is effectuation. In the innovation context, effectuation offers a major contribution by focusing on accomplishing innovation through an expanding cycle of resources. Goals emerge from courses of action and innovation is constructed by stakeholders integrated in an iterative conception [23]. In this way, the objective of this study was to interconnect effectuation to many approaches of innovation by identifying the differences and similarities between effectuation and frugal innovation, disruptive innovation, lean start-up, and design thinking.

Effectuation constitutes by itself the essence of innovation. Innovation is neither enacted nor organized. It only takes place when all the conditions are met: qualified people studying an innovative field and who will suddenly find their "eureka" and advance the knowledge (state of the art). Thus, effectuation joins the set of research works hailing a sociotechnical vision of innovation in which the notion of propagating an innovation is replaced by a construction of a network of all "components" involved in its success. In this context, innovation can be defined as discovering the goals and means of new innovation by interacting with interested stakeholders who are attracted to an entrepreneurial project as it unfolds [24].

#### 2.1.1. Linking Effectuation to Frugal Innovation

The issue of frugal innovation appears increasingly in publications on innovation, but also on sustainable development, because of the obvious implications for resource economics and the environmental and social aspects of sustainable development [25].

Frugal innovation attracts significant interest in the field of innovation [26]. The concept of frugal innovation encompasses two notions [27]: It is at the same time a means to innovate with limited means, and an objective to create simple and robust products destined (originally) to the emerging countries [28]. In what follows, the intention of the possible interconnection between effectuation and frugal innovation is focused on. Michaelis et al. [29] thought that a complementarity between frugal innovation and effectual logic is noticeable. Effectuation observes that entrepreneurs start with the means at their disposal, and not by considering those they could have.

#### 2.1.2. Linking Effectuation to Disruptive Innovation

The theory of disruptive innovation appeared in HARVARD BUSINESS REVIEW in 1995. This theory has proved to be a very effective tool for thinking about growth

based on innovation. Following the publication of "The Innovator's Dilemma" by Christensen [30], management researchers focused on innovations that "disrupt" consumers' demand models.

A new entrant creates an innovative product that initially is only in the interest of a niche market. According to classical criteria, this product is inferior to available products [3]. In the beginning, customers rejected it, but after any rapid attempts of amelioration, customers adopt it. To ameliorate disruptive product, disruptive innovation is associated with the creation of new sources of growth, typically the creation of new markets. The creation of a new market is the spirit of effectual logic [31]. New market creation is the result of the transformation process [32]. Human and artificial artifacts are a social artifact transformed to rules and standards [33].

### 2.1.3. Linking Effectuation to Design Thinking

Design thinking is an effective tool of innovation [34]. Effectuation and design thinking share the design concept. The interconnection between the effectuation logic and design thinking is the stakeholder's role in the innovation process [35]. Design thinking is aware of the risk of intervention, particularly of users, in the innovation process by focusing not on what the users say but on what the designer understands from the user requirements [36]. There is, in a way, a real need that the user is not able to express, and it is up to the designer to give birth to this need. There is a difference between developing a new product based on the indications of customers of the current products, or those of potential customers for the new product, and working with them to create the said product [37]. Effectuation confirms that the potential customers should not be consulted but co-create the product with them [38].

### 2.1.4. Linking Effectuation to Lean Start-Up

The concept of lean start-up, promoted by Ries [39] in his book, is inspired by the "lean" movement that is developed in the industry. One the principles of lean startup is "continue innovation". The core of lean start-up is starting quickly without waiting for the new product or service to be at the point because probably only the options for emergence, iteration, and refinement can be founded. The basic ingredients of lean start-up are "construction, measure and learn" [40].

This cycle englobes innovation: By building the project, measurement of the situations is needed, and this measure gives a vision of the errors, which allows learning and reconstruction by innovation. This idea of iterative and progressive development of startups is common with effectual logic [41]. This statement indicates that the spirit of entrepreneurship is not the perfect idea, but they continue the development of entrepreneurial activities.

### 2.2. Validation and Survey Methodology

To validate the studied framework, this research integrates the theories of entrepreneurial innovation and effectual logic, and offers an opportunity to interconnect some innovation types and approaches with effectual logic in order to identify the framework that reveals interactions between the constructs in question. The proposed methodology is based on the study of the publications that related effectuation to the four sectioned approaches of innovation.

Going further in this analysis, the bibliometrics tool "Harzing publish or perish version 7" was used. In the official website of the publish or perish tool, it is noted that the software is designed to decide which journals to submit to, to prepare for a job interview, to perform a literature review and survey, and to make a bibliometric research. Two criteria research were used. Indeed, results emanated from the following criteria: Scopus, publications type, journal title, last name of the first author, number of papers, and number of citations.

Two types of results were collected. Firstly, general results that deal with entrepreneurial innovation approaches were collected. Secondly, the association between effectuation and innovation approaches in title words research was selected.

## 3. Results

The importance of effectuation as a first specific theory in the field of entrepreneurship has led to a proliferation of publications. The development of effectuation research has been synchronized with a fostering of innovation research. In this stage, we proceeded to a bibliometric research via the publish or perish tool in order to identify the link between effectual logic and entrepreneurial innovation.

Table 1 asserts that acute volatility marks the research results. Utilizing, for instance, the keyword "design thinking" in the title, the software can list 1000 publications. Yet, by adding "effectuation" to "design thinking" as a keyword in the title we can enumerate just two publications. The combination between the "innovation" and "effectual" "effectuation" keywords identified 75 publications. These papers are detailed as follows: 18 publications combined "innovation" and "effectual" in title research and 57 associated "innovation" and "effectuation".

**Table 1.** Distribution of publications by research items.

| | Number of Publications | Number of Citations | Cites/Year | Cites/Paper | Author/Paper | Citation Years |
|---|---|---|---|---|---|---|
| "innovation" + "effectual" | 18 | 56 | 4.31 | 3.11 | 2.11 | 13 (2008/2021) |
| "innovation" + "effectuation" | 57 | 690 | 46 | 12.11 | 2.28 | 15 (2006/2021) |
| "Lean startup" | 740 | 10,579 | 622.29 | 14.3 | 1.59 | 17 (2004/2021) |
| "Lean startup" + "effectuation" | 2 | 30 | 15 | 15 | 1 | 2 (2019/2020) |
| "design thinking" | 1000 | 66,094 | 1101.57 | 66.09 | 2.36 | 60 (1961/2021) |
| "design thinking" + "effectuation" | 3 | 28 | 4 | 9.33 | 2 | 7 (2014/2021) |
| "frugal innovation" | 510 | 6662 | 416.38 | 13.06 | 2.16 | 16 (2005/2021) |
| "frugal innovation" + "effectuation" | 1 | 0 | 0 | 0 | 1 | 5 (2016/2021) |
| "disruptive innovation" | 530 | 23,951 | 1088.68 | 45.19 | 2.29 | 22 (1999/2021) |
| "disruptive innovation" + "effectuation" | 0 | 0 | 0 | 0 | 0 | 0 |

The appearance of effectuation and disruptive innovation is very close. The two approaches have a common area, which lies in the practice of entrepreneurship. Despite this, in the past 20 years, no publication has combined disruptive innovation and achievement. In another register, a unique publication combines "frugal innovation" and "effectuation". Frugal innovation as a word title research is mentioned in 510 publications that are cited 6662 times.

As we signaled above, the proposed analysis shows that an extant review for studying the effectuation and entrepreneurial innovation link gives volatile results. The most cited book is the revolutionary book of Eric Ries with 5767 citations (622.29 citations per year). The most cited publication in the research query "lean start-up" + "effectuation" is the paper of Ghezzi, A "Digital startups and the adoption and implementation of Lean Start-up Approaches: Effectuation, Bricolage and Opportunity Creation in practice". This paper was published in 2019 [42].

Tim Brown with his approach to design thinking is cited 5043 times. On the other hand and in association with effectuation "design thinking" and "effectuation", el Mansouri [41] is the most cited author with his paper intitled Comparing effectuation to discovery-driven planning, prescriptive entrepreneurship, business planning, lean startup, and design thinking. In the research results, the absence of publications that interconnect "disruptive innovation" to "effectuation" should be noted.

Table 2 presents the most cited publication according to the various used keywords. It should be noted that lean start-up and design thinking are the most innovation approaches cited in the literature, which are connected with effectuation.

**Table 2.** Distribution of the most cited publication.

| | Reference | Authors | Cites | Per Year | Year |
|---|---|---|---|---|---|
| "innovation" + "effectual" | [43] | AS Huff | 17 | 3.4 | 2016 |
| "innovation" + "effectuation" | [44] | H. Berends, M. Jelinek, I. Reymen, and R. Stultiëns | 321 | 45.86 | 2013 |
| "Lean startup" | [39] | Eric Ries | 5767 | 622.29 | 2011 |
| "Lean startup" + "effectuation" | [42] | Ghezzi, A. | 27 | 13.5 | 2019 |
| "design thinking" | [35] | Tim Brown | 5043 | 387.92 | 2008 |
| "design thinking" + "effectuation" | [37] | Y Mansoori and M Lackeus | 26 | 13 | 2019 |
| "frugal innovation" | [45] | Navi Radjou, Jaideep Prabhu | 810 | 90 | 2012 |
| "frugal innovation" + "effectuation" | [28] | S Fagbohoun | 0 | 0 | 2016 |
| "disruptive innovation" | [46] | CM Christensen, MB Horn, CW Johnson | 2632 | 263.2 | 2011 |
| "disruptive innovation" + "effectuation" | 0 | 0 | 0 | 0 | |

Table 3 shows that only three publications link design thinking to effectuation. From this perspective, the important element to signal is different themes: (1) corporate (2) business planning, and (3) early stage startups. The methodological nature of the papers overhead two qualitative papers and only one quantitative paper, which is inductive analysis. The qualitative research focused on the analytical comparison.

**Table 3.** Distribution of publications of design thinking related to effectuation.

| Publication Title | Years | Cites | Source | Authors and Co-Authors | Qualitative | Quantitative |
|---|---|---|---|---|---|---|
| Design thinking and effectuation in internal corporate venturing: an exploratory study | 2014 | 2 | alexandria.unisg.ch | T Abrell, M Durstewitz, F Uebernickel | Five internal corporate venturing projects in their early stage from idea to concept to a project | |
| Comparing effectuation to discovery-driven planning, prescriptive entrepreneurship, business planning, lean startup, and design thinking | 2019 | 26 | Small Business Economics | Y Mansoori, M Lackeus | Analytical visual comparison | |
| Uncovering the Link Between Effectuation and Design Thinking in Early Stage Startups | 2020 | 0 | Thesis/Aalto University | Varadarajan, Adithya | | Inductive analysis |

Regarding frugal innovation and as mentioned in Table 4, we can list a single article that links entrepreneurial innovation to effectuation. This paper was developed by Fagbohoun [28]. The author dealt with fablab case studies to investigate new approaches to practice innovation in an effectual context. The idea of fablab is very important to test good entrepreneurial practices in a practical context especially with the bird in hand principle. The idea of innovation with the available means is the core of effectual logic. The same idea is associated with frugal innovation.

**Table 4.** Distribution of publications of frugal innovation related to effectuation.

| Publication Title | Years | Cites | Source | Authors and Co-Authors | Qualitative | Quantitative |
|---|---|---|---|---|---|---|
| Frugal Innovation, Effectuation and FabLabs: Some Practices to Combine for a New Approach to Innovation | 2016 | 0 | Innovations | S Fagbohoun | Fablab cases | |

The two publications cited in Table 5, propose two different concepts. Mansouri's article associates lean startup with effectuation in a business planning logic. Conversely, Ghezzi treats lean startup compared to bricolage. In this publication, 227 digital startups were identified in mixed methods research.

**Table 5.** Distribution of publications of lean startup related to effectuation.

| Publication Title | Years | Cites | Source | Authors and Co-Authors | Qualitative | Quantitative |
|---|---|---|---|---|---|---|
| Comparing effectuation to discovery-driven planning, prescriptive entrepreneurship, business planning, lean startup, and design thinking | 2020 | 3 | Small Business Economics | Y. Mansoori | Analytical visual comparison | |
| Digital startups and the adoption and implementation of Lean Startup Approaches: Effectuation, Bricolage and Opportunity Creation in practice | 2019 | 27 | Technological Forecasting and Social | A. Ghezzi | | Mixed-methods research involving 227 digital startups |

## 4. Discussion

### 4.1. Linking Effectuation to Frugal Innovation

The resources of the entrepreneurs are rare, which is even the characteristic of entrepreneurship. In effectuation contexts, startups can be done without financial resources. It is a call to the imagination, to the improvisation and passage to the entrepreneurial action without or with a minimum of means [47].

It is an imagination of the re-use of a technology that has not been used or has been deemed useless. Frugal innovation also emphasizes flexibility and more generally the use of contingencies. This is also a point that the work emphasizes with the lemonade principle, which stresses that entrepreneurs take advantage of surprises, good or bad [25].

Despite the intersections between effectuation and frugal innovation, we note the presence of several differences. Firstly, effectuation postulates that the entrepreneur can start with practically nothing, as it does not target the entrepreneurial firm to an underprivileged environment [48]. The effect of the effectual cycle process allows starting small and finishing big. The effectual logic is not exclusive to grassroots populations as they do not have resources contrariwise, and international firms exploit effectuation principles. Effectuation is pertinent in emergent countries in starting entrepreneurial activities with limited resources, by guaranteeing performant results [49]. Secondly, effectuation put in the spotlight the social nature of innovation as an inverse of the creative and resourceful side put forward by frugal innovation. Creativity in an effectuation context is the result of cooperation with stakeholders [33].

### 4.2. Linking Effectuation to Disruptive Innovation

This similarity to effectuation logic is apparent and hides tacit differences. On the one hand, effectuators' reasons in terms of new products and services but disruptive innovators attempt to create a new simple offer, which focuses on some performance

criteria [50]. On the other hand, effectuation is the logic of co-creation of new products and services [51]. They are the result of the different stakeholders' commitment whose identity is not important and does not have the weight of accumulated means [52].

In the opposite way, the condition of the success of disruptive innovation is the link between the new product and organizational identity [53].

In other registers, the disruptive innovation market is limited, and they may remain limited or the market in an effectuation context is not limited since the effectuation principles are universal.

### 4.3. Linking Effectuation to Design Thinking

The difference between design thinking and effectuation is fundamental: in the design thinking case, the client gives an opinion without suffering the consequences, and it costs nothing to him. In the effectual case, customers engage in the development of products. This commitment can take various forms [54]. Table 6 illustrate the details of the difference between design thinking and effectuation.

**Table 6.** A comparison of the important aspects of effectuation and design thinking.

|  | **Design Thinking** | **Effectuation** |
| --- | --- | --- |
| Logic | General innovation | General innovation |
| Practice | User-centered | Crazy patchwork |
| Test | Iteration | Iteration |
| User | End users/stakeholders | Innovation with stakeholders |

The design thinking process is composed of six steps. In these steps, the spirit of effectuation is rooted. Iterations in this process are based on a user test. This idea is one of the most important conceptions of effectual logic, which evokes the crazy patchwork principle. The scope of design thinking is the same as effectuation: achieve general innovation. In this logic, the user is the core of the innovation process. Iteration is the axis of this process.

The innovative process in design thinking is connected to the spirit of innovation in effectuation logic by the design of a business and organization.

### 4.4. Linking Effectuation to Lean Startup

The affordable loss principle is a common zone between lean startup and effectuation [54] by investing what you are willing to lose in the evolution of the entrepreneurial process.

In another stage of analysis, there are significant differences between lean startup and effectuation. Effectual logic is universal, but lean startup is exclusively adapted to high-tech startups [55]. So, it is difficult to replicate it on industrial artifacts. Additionally, Ghezzi and Cavallo [56] stated that for lean startup, the iteration is an interactive exchange with the market in order to adapt the product. On the other side, effectuation is understood as iteration not as the improvement of the product. Table 7 illustrate the details of the difference between lean startup and effectuation.

**Table 7.** A comparison of the important aspects of effectuation and lean startup.

|  | **Lean Startup** | **Effectuation** |
| --- | --- | --- |
| Logic | High-tech innovations for startups | General innovation |
| Practice | Customer oriented | Crazy patchwork |
| Test | Metrics | Iteration |
| User | End users/stakeholders | Innovation with stakeholders |

The innovative process in lean startup is connected to the spirit of innovation in effectuation logic by the problem solution fit.

This research shows that effectuation and innovation intersect from a theoretical and practical point of view. The innovation practices dealt with in this research are interconnected with the achievement in the literature of significant volatility. We contribute in this research to a better understanding of the nature of innovation and effectuation. It is core as a design science finds its links with tools, typically the design thinking and lean startup. The major restriction is to proceed in this search using only google scholar on the publish and perish tool. In future research, diversification of the databases is a preference in order to discus a framework based on effectuation and innovation as design science.

## 5. Conclusions

Effectuation and innovation are two sides of the same coin. Effectuation starts gaining visibility outside academic circles because it is vital to understand how entrepreneurs think and act in their creative process. Effectuation is the way to explain innovation in entrepreneurship research. This paper is a trial to interconnect effectuation to different types and approaches of innovation. We sectioned four items in relation to the spirit of effectuation logic. We found that the four types and approaches of innovation can be interconnected with effectuation. Going further in this study publish or perish', we used the bibliometrics tool 'Harzing. To validate the study, a bibliometrics tool, named 'Harzing publish or perish', was used. The main finding of this research affirms that the two most linked innovation approaches to effectuation are "lean start-up" and "design thinking", compared to "frugal innovation" and "disruptive innovation". In an entrepreneurial innovation context, design thinking and lean start-up are flexible tools that can stimulate and validate the effectual cycle. These are interesting results, but in-depth causal and synthetic studies are still needed. In addition, future studies, other approaches, and models of innovation can be studied in the orbit of effectuation theory.

**Author Contributions:** Conceptualization, F.G., Y.B. and W.H.; methodology, F.G., W.H. and A.M.A.; validation, W.H., Y.B. and A.M.A.; formal analysis, F.G., and W.H.; investigation, F.G.; resources, F.G. and A.M.A.; data curation, F.G. and Y.B.; writing—original draft preparation, F.G.; writing—review and editing, W.H. and A.M.A.; visualization, Y.B. and A.M.A.; supervision, W.H. and Y.B.; project administration, W.H. and A.M.A.; funding acquisition, A.M.A. All authors have read and agreed to the published version of the manuscript.

**Funding:** This research was supported and funded by Taif University Researchers Supporting Project number (TURSP-2020/229), Taif University, Taif, Saudi Arabia.

**Institutional Review Board Statement:** Not applicable.

**Informed Consent Statement:** Not applicable.

**Data Availability Statement:** Data is contained within the article.

**Acknowledgments:** This research was supported by Taif University Researchers Supporting Project number (TURSP-2020/229), Taif University, Taif, Saudi Arabia. Firstly, the authors are grateful for this financial support. Secondly, the authors would like to thank the editor and the three anonymous reviewers, whose insightful comments and constructive suggestions helped us to significantly improve the quality of this paper.

**Conflicts of Interest:** The authors declare no conflict of interest.

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
