# Peer review of "Linking Entrepreneurial Innovation to Effectual Logic"

_sustainability, doi:10.3390/su13052626_

Round 1

Reviewer 1 Report

  1. Introduction

The purpose of the paper is clearly stated.

The problem is well defined.

The case is made that the problem is significant.

The research questions are clear.

The title, abstract, and introduction could include more information about the period of study.

  1. Literature Review

The literature is recent and relevant.

The literature review acknowledges the depth and breadth of investigation in the field.

  1. Methods

The methods used are clearly explained and justified.

The methods are sufficient to answer the research questions.

  1. Findings

The results are clearly related to the data (including text and numeric).

Figures, tables, and other graphic displays are understandable, and their primary findings are discussed in the text.

The arguments are presented with sufficient logical consistency.

  1. Discussion and Implications

The findings are not clearly discussed in relation to the literature review and research questions.

Implications for theory, policy, and practice are not explored in detail.

The strengths and limitations of the study are not adequately discussed.

Author Response

The authors would like to thank the editor and the three anonymous reviewers, whose insightful comments and constructive suggestions helped us to significantly improve the quality of this paper. All modifications in the manuscript are mentioned in red colour. 

Reviewer 2 Report

  1. Research questions should be more precise.
  2. The argument for the four "types and approaches" of innovation is too weak (basically no reference behind). Why authors did not take Schumpeter's categorization?
  3. Research questions are not exactly answered in the final part of the paper.
  4. Table 2. in column "Publication" and in rows "frugal innovation", "disruptive innovation" and "disruptive innovation" + "effectuation" contains numbers which are not appropriate here.

Author Response

(The authors gave the same response as above.)

Reviewer 3 Report

The authors aim to link entrepreneurial innovations to effectual logics. By doing so, they choose to use the Publish or Perish tool for their research purpose. I believe that the approach sounds interesting, but I have the methodological concern that the tool has not been explicitly explained. On its website, I read it that it is a tool for for preparing for job interviews etc. so the authors need to justify their selection of the tool.

Furthermore, and this is something associated with the keywords I suppose, some of the most important and most cited papers on the topic were surprisingly neglected.

For example, "Effectuation, innovation and performance in SMEs: an empirical study" by Roach et al., cited 93 times,

"Effectuation or causation as the key to corporate venture success? Investigating effects of entrepreneurial behaviors on business model innovation and venture performance" by Futterer et al., cited 81 times,

just to give you an idea. I think that any literature review which neglects the most cited papers deserves improvement, and I hope that these comments can help the authors improve their analysis.

Author Response

(The authors gave the same response as above.)

Round 2

Reviewer 3 Report

Li. 258; the tool’s name is wrong as it is publish or perish (not and).

Apart from this minor issue I am satisfied with the new version.

Author Response

Tank you for your precision.

You are right.

The tool’s name are corrected as your remark